# Long-Term Impact of Direct-Acting Antivirals on Liver Fibrosis and Survival in HCV-Infected Liver Transplant Recipients

**DOI:** 10.3390/v15081702

**Published:** 2023-08-07

**Authors:** Martina Gambato, Chiara Manuli, Erica N. Lynch, Sara Battistella, Giacomo Germani, Marco Senzolo, Alberto Zanetto, Alberto Ferrarese, Alessandro Vitale, Enrico Gringeri, Umberto Cillo, Patrizia Burra, Francesco Paolo Russo

**Affiliations:** 1Multivisceral Transplant and Gastroenterology Unit, Department of Surgery, Oncology and Gastroenterology, Azienda Ospedale-Università Padova, Via Giustiniani 2, 35100 Padova, Italy; chiara.manuli@gmail.com (C.M.); ericanicolalynch@gmail.com (E.N.L.); sarabattistella93@gmail.com (S.B.); giacomo.germani@gmail.com (G.G.); marco.senzolo@hotmail.com (M.S.); albertoferrarese17@gmail.com (A.F.); burra@unipd.it (P.B.); 2Department of Surgery, Oncology and Gastroenterology, University of Padova, 35100 Padova, Italy; alessandro.vitale@unipd.it (A.V.); enrico.gringeri@unipd.it (E.G.); cillo@unipd.it (U.C.); 3Hepatobiliary Urgery and Liver Transplantation, Department of Surgery, Oncology and Gastroenterology, Azienda Ospedale-Università Padova, 35100 Padova, Italy

**Keywords:** hepatitis C virus, DAAs, liver transplantation

## Abstract

(1) Background: Little is known about the long-term impact of sustained virological response (SVR) on fibrosis progression and patient survival in liver transplantation (LT) recipients treated with direct-acting antivirals (DAAs). We investigated liver fibrosis evolution and patient survival in hepatitis C virus (HCV)-infected patients receiving DAAs after LT. (2) Methods: All consecutive HCV-infected patients treated with DAAs after LT between May 2014 and January 2019 were considered. The clinical and virological features were registered at the baseline and during the follow-up. The liver fibrosis was assessed by liver biopsy and/or transient elastography (TE) at the baseline and at least 1 year after the end of treatment (EoT). (3) Results: A total of 136 patients were included. The SVR12 was 78% after the first treatment and 96% after retreatment. After the SVR12, biochemical tests improved at the EoT and remained stable throughout the 3-year follow-up. Liver fibrosis improved after the SVR12 (*p* < 0.001); nearly half of the patients with advanced liver fibrosis experienced an improvement of an F ≤ 2. The factors associated with lower survival in SVR12 patients were the baseline platelet count (*p* = 0.04) and creatinine level (*p* = 0.04). (4) Conclusions: The long-term follow-up data demonstrated that SVR12 was associated with an improvement in hepatic function, liver fibrosis, and post-LT survival, regardless of the baseline liver fibrosis. The presence of portal hypertension before the DAAs has an impact on patient survival, even after SVR12.

## 1. Introduction

The advent of direct-acting antivirals (DAAs) greatly improves viral eradication rates—which are higher than 90% in most HCV-infected populations—including patients with cirrhosis [1], leading to a significant reduction in the number of patients who undergo liver transplantation (LT) for HCV-related liver disease [2,3]. In fact, DAAs have been reported to improve liver function, clinical condition, and survival rates in different series of non-transplanted patients [4,5,6,7,8]. Different DAA regimens have been proven highly effective and safe in liver transplant recipients, with sustained virological response (SVR) rates above 90% in both trials [9,10] and real-life series [11,12,13,14,15]. The safety and efficacy of DAAs after LT are widely accepted, but only a few studies have described the long-term impact of viral eradication on liver function and survival after LT [16,17,18]. Mauro et al. [19] reported data collected from 112 HCV-infected LT recipients who achieved SVR and whose degree of fibrosis and portal hypertension were evaluated over a 1-year period using both invasive and non-invasive methods. SVR after LT was associated with a regression in the biopsy-proven liver fibrosis stage and portal hypertension, although the latter appeared to improve less significantly in patients with advanced liver disease at the baseline [19]. In addition, after the introduction of DAAs, patient [2,3] and graft survival [20,21] after LT for HCV-related liver disease significantly improved, becoming similar to the survival rates of other indications [2].

Hereby, we aimed to evaluate the long-term impact of viral eradication on fibrosis progression and patient survival after LT in HCV-infected recipients, reporting the efficacy and safety data of DAAs in this population. In addition, we investigated hepatocellular carcinoma (HCC) recurrence after LT in recipients treated with DAAs after transplantation.

## 2. Materials and Methods

### 2.1. Patient Population and Clinical Data

In this retrospective cohort study, we prospectively collected and analyzed data from all consecutive HCV-infected LT recipients who received DAA treatment after LT between May 2014 and January 2019 and who were included in the NAVIGATORE web-based platform or in the DAAs’ compassionate use program in Italy. Patients who prematurely discontinued treatment and those who had not been evaluated at least 12 weeks after the end of treatment (EoT) were excluded.

Variables recorded before DAA initiation included: demographics, immunosuppressive regimen, previous HCV treatment and response, blood cell count and hemoglobin levels, liver tests, the HCV-RNA genotype and serum level [real-time reverse transcriptase polymerase chain reaction (real-time RT-PCR) with a lower limit of detection 12 IU/mL], and the hepatitis B surface antigen. Baseline staging of liver disease was performed by liver biopsy or transient elastography (TE) (Fibroscan, Echosens, Paris, France). At least one of the following criteria defined the presence of cirrhosis: biopsy-proven fibrosis stage METAVIR = 4, the presence of esophageal varices, or liver stiffness measurement (LSM) ≥ 12 kPa. Cut-off values used for liver stiffness were as follows: F0–F1 < 7 kPa, F2 ≥ 7, and <9 kPa, F3 ≥ 9, and <12 kPa, F4 ≥ 12 kPa. In patients with liver cirrhosis, liver disease severity scores [model for end-stage liver disease (MELD) and Child–Turcotte–Pugh (CTP) scores] were calculated. Patients underwent clinical and laboratory assessments (liver function tests, blood cell count, ALT, AST, serum creatinine, and HCV-RNA levels) every 4 weeks during the treatment period and at 3, 6, 12, 24, and 36 months after the EoT. They continued the post-transplant follow-up thereafter, according to our protocol. After viral eradication, transient elastography was performed at least 1 year after the EoT.

### 2.2. The NAVIGATORE Database

Since March 2015, all HCV-infected solid organ transplant recipients have been eligible for DAA treatment. Data relative to treatments have been recorded on the “NAVIGATORE” database, a web-based platform on which clinical and virological data of all HCV-infected patients treated with DAAs in the Veneto region (Italy) are recorded prospectively. All the 26 clinical centers authorized to prescribe DAAs by the Veneto Regional Governing Board are involved in the project. All patients received DAA treatment based on available DAA regimens and according to national and European guidelines. 

### 2.3. Compassionate Use Program in Italy

Between June 2014 and January 2015, patients were treated in the context of the Italian sofosbuvir compassionate use program. For each patient, the treatment had to be approved by the local ethics committee, and all patients provided written informed consent. Criteria of inclusion in the compassionate use program in Italy were: (1) patients with decompensated cirrhosis on the waiting list for LT and (2) patients with severe HCV recurrence after LT (fibrosing cholestatic hepatitis or METAVIR fibrosis stage > F2). Post-transplant patients were treated with sofosbuvir (SOF) 400 mg (one tablet per day) in combination with ribavirin (1.000 to 1.200 mg/day) for 24 weeks, according to tolerance and medical decision.

### 2.4. Clinical Outcomes

(1)Efficacy and safety data:

Treatment efficacy was measured as SVR12, defined as undetectable HCV-RNA by a quantitative RT-PCR with a limit of detection of 12 IU/mL 12 weeks after the EoT. 

Patients were monitored for clinical and laboratory evidence of adverse events at each study visit during the treatment period. Any event that occurred during treatment was reported, regardless of the need for prescription medication or a DAA dose reduction.

(2)Fibrosis evolution after SVR:

Changes in liver fibrosis after LT were classified according to the liver stiffness value as follows:-Fibrosis improvement: reduction in METAVIR class from F3–4 to F0–2.;-Fibrosis worsening: increase in METAVIR class from F0–2 to F3–4;-Fibrosis stabilization: unchanged METAVIR class.
(3)Patient and graft survival:

For all patients, the last follow-up control visit was considered to determine patient survival and re-transplantation. Cause of death was also recorded when available. 

(4)HCC recurrence after LT:

Patients with an HCC before LT who received DAAs after LT were analyzed separately. Features of HCCs in explant liver specimens were recorded. According to our protocol, patients underwent radiological screening for HCC recurrence every 6 months after LT by abdominal and pulmonary CT scans for the first 5 years after LT and abdominal ultrasound thereafter. 

### 2.5. Statistical Analysis

Groups were compared using a *t*-test or Mann–Whitney test for continuous variables and chi-square or Fisher tests for categorical variables. A Wilcoxon test was used to compare paired variables. Patient survival was evaluated using the Kaplan–Meier method and compared using the log-rank test or the univariate and multivariate Cox regression models. 

All differences and associations were considered significant at a 2-sided *p*-value of <0.05. Statistical analyses were performed with SPSS, version 25 (2017, IBM Corp, Armonk, NY, USA).

The study has been conducted according to the ethical guidelines of the 2013 Declaration of Helsinki and approved by the local ethics committee.

## 3. Results

### 3.1. Baseline Characteristics of Patient Population

Overall, 143 HCV-infected LT recipients were treated with DAAs between April 2014 and January 2019. Among them, seven patients were excluded from the analysis: two patients for treatment discontinuation and five patients for insufficient follow-ups to determine SVR12. 

Overall, 136 patients treated with DAAs after LT were included in the study. The demographic, virological, and clinical characteristics of patients at baseline are depicted in Table 1. 

Two patients were treated after a combined liver–kidney transplantation. The liver fibrosis stage at the baseline was F4 in 41% of patients (n = 56), F3 in 23% (n = 31), F2 in 22% (n = 30), and F0–1 in 14% (n = 19). Among cirrhotic patients, 64% presented compensated liver disease with CTP class A (n = 36), and the median MELD score was 8 (IQR 7–12). 

The main immunosuppressant was FK in one hundred and two patients (76%), cyclosporin in twenty-three patients (17%), mTOR inhibitors in three patients (2%), and other drugs in six patients (5%). Additional immunosuppressive drugs were used as follows: mycophenolate in thirty-six patients (26%), steroids in seventeen patients (12%), and azathioprine in eight patients (6%). 

The median follow-up after DAA treatment initiation was 51 months (IQR 41–56). The median time between LT and DAA treatment was 62 months (IQR 16–122). In patients with cirrhosis before DAAs, the median time from LT was 78 months (IQR 26–138). 

### 3.2. Efficacy and Safety of DAA Treatment after LT

The SVR12 rate after the first DAA regimen was 78.7% (n = 107). Among the twenty-nine relapsers, twenty-seven patients received a second DAA treatment [SOF/LDV + RBV in sixteen patients (59%), SOF/DCV ± RBV in six patients (22%), SOF/VEL/VOX in one patient (4%), and SOF/SMV + RBV in four patients (15%)], and the other two patients died before receiving a second DAA regimen (one from hepatic failure and one from an unknown cause). Twenty-two patients (81%) responded to a second DAA treatment. Among the five relapsers to the second treatment, one patient received a third DAA regimen (SOF/VEL/VOX) with SVR and four patients died (two patients from hepatic failure and two patients from recurrent HCC). Overall, the SVR12 rate was 96%, including responders to the second DAA regimen. 

Overall, 27% (n = 36) of patients reported the occurrence of adverse events (AEs) during the treatment period. The details are provided in Appendix A. Twenty-eight percent (n = 37) of patients needed adjustment of the RBV dose. No patient experienced graft rejection.

### 3.3. Biochemical Tests after SVR

In LT recipients who achieved SVR, the platelet count increased significantly at the EoT (*p* < 0.001) and then remained stable at SVR12, at SVR24, and at year 1, 2, and 3 after the EoT (*p* = 0.06, *p* = 0.74, *p* = 0.46, *p* = 0.18, and *p* = 0.37). The same result was observed in patients with liver cirrhosis. ALT levels decreased significantly at the EoT (*p* < 0.001) and from the EoT to SVR12 (*p* = 0.001) and then remained stable at SVR24 and at year 1, 2, and 3 after the EoT (*p* = 0.42, *p* = 0.84, *p* = 0.65, and *p* = 0.23). In cirrhotic patients, ALT levels decreased significantly at the EoT (*p* < 0.001) and from the EoT to SVR12 (*p* = 0.008) and remained stable thereafter (*p* = 0.5, *p* = 0.57, *p* = 0.05, and *p* = 0.57). 

Overall, creatinine levels decreased at the EoT (*p* = 0.04) and remained stable throughout the 3 years of follow-up (*p* = 0.07, *p* = 0.17, *p* = 0.13, *p* = 0.35, and *p* = 0.3). Among patients with impaired renal function at the baseline (eGFR < 60/mL/1.73 m^2^) (n = 30), creatinine levels decreased significantly at the EoT (*p* = 0.02) and were stable during the remaining follow-up (*p* = 0.53, *p* = 0.96, *p* = 91, *p* = 0.19, and *p* = 0.67). 

In cirrhotic patients, the APRI score decreased significantly at the EoT (*p* < 0.001) and remained stable at year 2 and 3 after the EoT (*p* = 0.41, *p* = 0.62).

### 3.4. Liver Stiffness Evolution in LT Recipients after SVR

Among patients who achieved SVR (n = 131), 77 patients had paired liver stiffness measurements [LSM] at the baseline and between the first and the fourth year after the EoT. The follow-up liver stiffness measurement was performed at a median time of 33 months (IQR 19.5–46.5) after the EoT. Thirty-one percent (n = 24) of patients showed fibrosis improvement, 61% (n = 47) an unchanged liver fibrosis stage, and 8% (n = 6) showed fibrosis worsening. Overall, liver stiffness decreased significantly from a median value of 11.9 kPa (IQR 8–16.4) to a median value of 7.9 kPa (IQR 5.9–14.2) (*p* = 0.003). 

The distribution of liver fibrosis at the baseline and during the follow-up in patients who achieved SVR is represented in Figure 1.

Considering the fibrosis stage at the baseline among patients with paired LSM, patients with F3–4 fibrosis at the baseline experienced an overall reduction in liver stiffness that significantly decreased from a median value of 14 kPa (IQR 11.4–19.6) to a median value of 9.7 kPa (IQR 7–15.7) (*p* < 0.001). In 25 out of 49 patients (51%), the fibrosis stage improved from the F3–4 stage to the F0–2 stage, whereas in 24 out of 49 patients (49%) the liver fibrosis stage remained unchanged.

Overall, in 22 out of 28 patients with F0–2 fibrosis at the baseline (79%), liver stiffness remained stable during the follow-up (*p* = 0.6). In six out of twenty-eight (21%), the fibrosis stage worsened from F0–2 to F3–4.

Among the six patients who experienced a worsening of liver fibrosis, three patients suffered from metabolic syndrome and one developed post-transplant de novo autoimmune hepatitis; the other two patients had no risk factors for liver disease progression. 

### 3.5. Survival Analysis

Overall, the cumulative patient survival at 1, 2, 3, and 4 years after the DAAs was 99%, 96%, 92%, and 86%, respectively. For patients with a baseline liver cirrhosis cumulative survival at 1, 2, 3, and 4 years after the DAA treatment, it was 98%, 93%, 81%, and 72%, respectively, which was not significantly different from the non-cirrhotic population, who showed a cumulative survival of 100%, 95%, 85%, and 79%, respectively (*p* = 0.16) (Figure 2a). 

In the univariate analysis, the factors associated with lower survival were relapse after DAA treatment (*p* < 0.001), the platelet count < 50,000/mm^3^ (*p* = 0.003), a higher creatinine level at the baseline (*p* = 0.03), and the presence of liver decompensation at the baseline (*p* = 0.046). In the multivariate analysis, relapse following DAA treatment was confirmed as the only independent risk factor for mortality (*p* = 0.008) (Figure 2b). 

After excluding the five relapsers who died before initiating a second course of DAAs, the only two factors that were found to be associated with lower survival in the univariate and multivariate analyses were platelet count < 50,000/mm^3^ (*p* = 0.04) and higher creatinine levels at the baseline (*p* = 0.03) (Table 2).

Patients who achieved SVR with the first DAA regimen and patients who achieved SVR after multiple DAA treatments had similar survival (*p* = 0.59).

The cause of death was known for 20 out of 22 patients. Seven patients (35%) died from HCC progression, six patients (30%) from liver disease decompensation, four patients (20%) from the progression of extrahepatic de novo neoplasms (glioblastoma, acute myeloid leukemia, post-transplant lymphoproliferative disorder, and thymoma), one patient (5%) from chronic rejection, one patient (5%) from sepsis, and one patient (5%) from a cardiovascular event. 

### 3.6. HCC Recurrence

Overall, the LT indication was HCV-related HCC in 65 patients (48%). Among them, two patients developed HCC recurrence before the DAA treatment and eight patients (12%) at a median time of 45 months after the LT (IQR 26–80) and a median time of 21 months (IQR 10.5–39) after DAA initiation. One patient who had been transplanted for HCV-related cirrhosis developed de novo post-transplant HCC 62 months after the LT.

Details about the histology of the explanted liver were available for 57 patients (88%) and are reported in Table 3. For the liver explant, 24 HCC (42%) were Milan-in, 21 HCC (37%) were Milan-out, and 12 did not show residual HCC (21%). The majority of HCC at the explant had G2 grading (n = 33, 58%). Sixteen patients (28%) presented microvascular invasion. Among the eight patients who experienced HCC recurrence after DAA treatment, six patients (75%) presented G2 HCC, five (62%) fulfilled the Milan criteria, and three (33%) showed macrovascular invasion in the explant liver pathology.

## 4. Discussion

Since the introduction of DAAs, several studies have documented their high safety and efficacy in LT recipients [9,10,11,12,13,14,15]. In our series, the SVR rate following the first DAA regimen was 78% lower than those reported in most studies [10,12,13]^.^ However, after retreatment, 96% of patients achieved SVR. In this regard, it should be noted that 44% of patients in our study had been treated with SOF/RBV as part of the compassionate use program and achieved a lower SVR rate due to the suboptimal regimen, which was used and to the fact that most patients had cirrhosis.

With respect to AEs, we confirmed the excellent safety profile of DAAs in transplanted patients, as they were well tolerated, with no serious AEs occurring during the treatment period. In our cohort, mild and moderate AEs were reported in patients who received RBV. 

Viral eradication is associated with liver inflammation resolution that may lead to the reduction of liver fibrosis, which represents the main purpose of the eradicating therapy [22]. Most studies reported the short-term outcomes of HCV-infected liver recipients, demonstrating an improvement in liver inflammation, fibrosis [17,23,24], and disease severity [15,25]. Mauro et al. [19] reported both histological data and portal pressure measurements 1 year after the EoT in 60 HCV-infected LT recipients treated with DAAs who achieved SVR. SVR was associated with biopsy-proven fibrosis regression regardless of the baseline liver fibrosis stage, even though the fibrosis improvement was less frequent in patients with baseline liver cirrhosis and clinically significant portal hypertension; cirrhosis regression was also associated with portal pressure reduction. In addition, they observed a significant reduction in liver stiffness, demonstrating that liver stiffness measurements were highly accurate in detecting or discarding the presence of advanced liver fibrosis 1 year after SVR. Martini et al. [16] reported data from 120 LT recipients treated with SOF + RBV, showing a reduction in transaminase levels early after treatment initiation, as well as an improvement in bilirubin, the platelet count, and albumin levels at SVR24, which did not change significantly up to 1 year of the follow-up. Moreover, they compared the LSM in 78 LT recipients with advanced baseline liver fibrosis after DAA treatment and observed a significant reduction at SVR24 and a further improvement at SVR48 in F4 patients. 

Similarly, in our study, long-term follow-up evaluations after DAA treatment showed that the transaminase levels and platelet count improved at the EoT and maintained a stable trend up to 3 years after the EoT. Moreover, we confirmed a significant decrease in liver stiffness after successful DAA treatment. For the first time, we were able to report a long-term improvement in liver stiffness in patients with advanced liver fibrosis at the baseline (64%). As it has been suggested that the changes in liver stiffness occurring early after DAA treatment might be related to the resolution of hepatic inflammation rather than to the regression of liver fibrosis, we only considered liver stiffness measurements performed at least 1 year after the EoT, which may more accurately correlate with the actual extent of residual hepatic fibrosis and have a better prognostic value. Most importantly, half of the patients with advanced fibrosis at the baseline experienced regression in fibrosis stages < 2, which led to a change in the management and surveillance of these patients. Patients who did not experience fibrosis regression might have reached a point of no return in the progression of their disease or might have been exposed to other risk factors for liver disease. 

Recently, Liu et al. [26] published their experience with a large cohort of LT recipients who obtained SVR after interferon-based regimens and after DAA treatment, demonstrating that SVR associated with both types of treatment was associated with improved or halted fibrosis progression. 

The only study [27] that compared liver biopsies performed at the baseline and after antiviral treatment within a long-term follow-up (3 years) included 12 patients with baseline liver cirrhosis and confirmed our result, demonstrating a significant improvement in hepatic fibrosis and inflammation. They also monitored patients for changes in kidney function potentially caused by DAA regimens containing sofosbuvir and observed that kidney function was not affected by antiviral treatment. Renal function remained stable without declining within a 1-year follow-up. Similarly, in our study we observed that kidney function was not affected by DAAs; patients experienced a slight decrease in creatinine levels after the EoT, which remained unchanged thereafter, even in patients with severe kidney injury at the baseline. Given the correlation between HCV infection and kidney disease, we confirmed that patients with impaired renal function may benefit from viral eradication. 

Different studies have compared the patient and graft survival of HCV-infected LT recipients before and after the introduction of DAAs, demonstrating a significant survival gain in the DAAs’ era [2,21,28]. Vukotic et al. [29] reported data from LT recipients with baseline cirrhosis or fibrosing cholestatic hepatitis treated in different Italian centers within the compassionate use program, describing poor survival in patients with decompensated liver disease before antiviral treatment. Similarly, in our study, five relapsers with advanced disease died because of hepatic decompensation or HCC progression before they could receive a second DAA regimen. However, in patients who achieved SVR after a second treatment, advanced fibrosis at the baseline was not associated with lower survival, suggesting that after viral eradication patients had a lower risk of death due to hepatic decompensation. Nevertheless, patients with a platelet count < 50,000/mm^3^, which was considered a surrogate for significant portal hypertension, experienced lower survival, confirming that these patients might be less likely to benefit from DAA treatment [18]. In addition, patients with higher creatinine levels at the baseline had lower long-term survival; this finding is more difficult to interpret due to the multiple factors that can affect kidney function in transplanted patients.

Data from large European [28] and American [21] transplant databases comparing the post-transplant survival of patients before and after the introduction of DAAs demonstrated a survival benefit for patients with HCV-related liver disease, which could only be attributed to antiviral treatment. A report from the European Liver Transplant Registry (ELTR) [2] confirmed this trend and found that for patients transplanted for HCV-related HCC, liver decompensation decreased as a cause of death, whereas HCC recurrence remained stable. In our cohort, the most frequent cause of death in SVR patients was HCC recurrence. We were not able to demonstrate that HCC recurrence was related to antiviral therapy due to the small sample size and considering the time between tumor recurrence and the DAAs (the majority was >1 year after the DAA). Regarding HCC characteristics, we did not find any association between HCC morphology in the explant pathology and HCC recurrence. Our findings are in line with previous findings. A recent international multicenter study [30] compared the risk of HCC recurrence in LT recipients treated with DAAs before and after LT and untreated patients, showing that DAA treatment did not increase the risk of HCC recurrence after LT. These results were confirmed by Piñero et al. [31], who investigated the risk of progression and recurrence of HCC in a large cohort that included patients treated with DAAs after LT. 

One of the main limitations of our study is the retrospective design; for this reason, the follow-up LSMs were not available for all patients and were performed at different timepoints. However, we only considered the LSM, which had been performed at least 1 year after the EoT. This is the only study evaluating the long-term impact of DAAs on liver fibrosis and survival in HCV-infected liver transplant recipients. In addition, we reported efficacy and safety data, which are useful for the management of HCV-positive recipients and patients who receive an HCV-positive organ.

## 5. Conclusions

In conclusion, our findings confirmed that viral eradication with DAAs after LT improves liver function and patient survival after a long-term follow-up, even in cirrhotic patients. In addition, our study showed that LT recipients with portal hypertension before treatment were less likely to benefit from DAAs and experienced lower survival rates. 

## Figures and Tables

**Figure 1 viruses-15-01702-f001:**
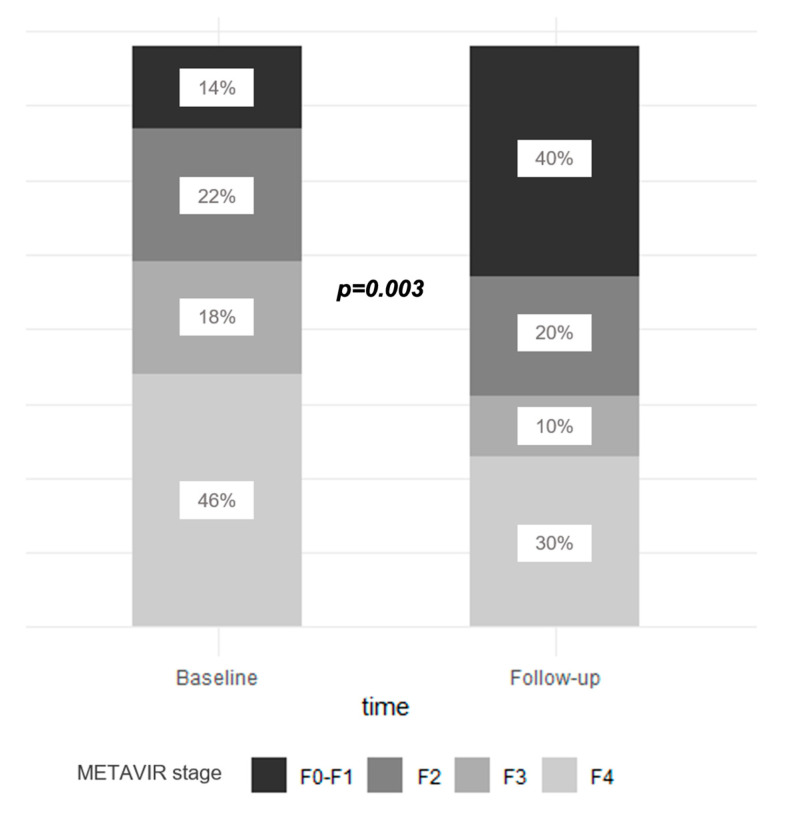
Liver fibrosis stage in patients with paired liver fibrosis assessment (n = 78) at the baseline and during the follow-up after sustained virological response (SVR).

**Figure 2 viruses-15-01702-f002:**
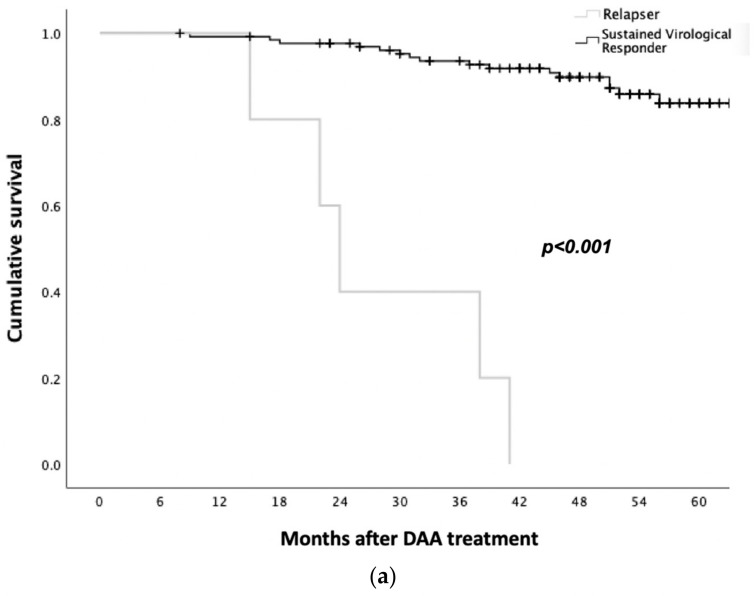
(**a**) Survival of patients grouped by response to antiviral treatment. (**b**) Survival of patients grouped by baseline liver fibrosis.

**Table 1 viruses-15-01702-t001:** Baseline characteristics of patient population.

N = 136	N (%) Median (IQR)
Sex (male)	108 (79%)
Age (years)	58 (52–66)
HCV genotype	
- 1	102 (75%)
- 2	11 (8%)
- 3	17 (13%)
- 4	6 (4%)
Log HCV-RNA (UI/mL)	6.2 (5.7–6.7)
Compassionate use program/Navigatore	60 (44%)/76 (56%)
DAA regimen	
- SOF + RBV	81 (60%)
- SOF/DCV ± RBV	5 (4%)
- SOF/LDV ± RBV	33 (24%)
- SOF/SMV ± RBV	13 (9%)
- SOF/VEL ± RBV	3 (2%)
- SOF/VEL/VOX	1 (1%)
Ribavirin use	125 (92%)
Treatment duration	
- 24 weeks	94 (69%)
- 12 weeks	42 (31%)
Time between transplant and DAA treatment (months)	62 (16–122)
LT indication	
- HCC	65 (48%)
- Decompensated cirrhosis	71 (52%)
Main immunosuppression	
- FK	102 (76%)
- CsA	23 (17%)
- mTOR-i	3 (2%)
- Other	6 (5%)
METAVIR stage before DAA *	
- F0–1	19 (14%)
- F2	30 (22%)
- F3	31 (23%)
- F4	56 (41%)
Platelet count (×10^9^/L)	121 (84–157)
Platelet count < 50 × 10^9^/L	
ALT (IU/L)	65 (41–105)
Total bilirubin (mg/dL)	0.72 (0.5–1.11)
INR	1.06 (1.1–1.12)
Albumin (g/L)	39 (36–41)
Creatinine (mg/dL)	0.96 (0.81–1.18)
MELD score (in patients with cirrhosis)	8 (7–12)
Child–Pugh class (in patients with cirrhosis)	
- A	36 (80%)
- B	9 (20%)

* Baseline liver fibrosis stage was evaluated with hepatic elastography in 78% patients, liver biopsy in 21%, and the presence of signs of portal hypertension in 1%.

**Table 2 viruses-15-01702-t002:** Univariate and multivariate survival analyses in patients who achieved sustained virological response (SVR). *HR: Hazard Ratio*. ***: statistically significant*.

	Univariate Cox			Multivariate Cox		
	HR	95% CI	*p*-Value	HR	95% CI	*p*-Value
Age (years)	1.042	0.99–1.09	0.116			
Gender (male)	4.971	0.65–37–54	0.120			
Advanced fibrosis at baseline	2.094	0.59–7.41	0.252			
LT indication (HCC)	1.265	0.48–3.28	0.630			
Time between LT and DAAs	0.998	0.99–1	0.530			
Platelet count < 150,000/mm^3^ at baseline	2.372	0.53–10.56	0.257			
Platelet count < 50,000/mm^3^ at baseline	3.768	0.85–16.73	0.081	5.08	1.09–23.61	0.038 **
Albumin level at baseline	0.946	0.87–1.02	0.165			
INR at baseline	0.946	0.03–25.53	0.946			
Creatinine level at baseline	3.918	1–15.3	0.049 **	4.942	1.2–20–37	0.027 **
Liver decompensation at baseline	2.019	0.51–7.97	0.316			

**Table 3 viruses-15-01702-t003:** Explant histologic features in patients transplanted for HCV-related HCC (explant features available in n = 57) and those with HCC recurrence after DAA.

Median (IQR) N (%)	Overall n = 57 *	Patients with HCC Recurrence after DAA N = 8 **	Patients without HCC Recurrence after DAA N = 49 ***	*p*-Value
Vascular invasion	16 (28%)	3 (38%)	13 (24%)	0.522
Milan criteria:				0.237
- In	24 (42%)	5 (62%)	19 (39%)	
- Out	21 (37%)	3 (38%)	18 (37%)	
- Absence of HCC	12 (21%)	0 (0%)	12 24%)	
HCC grading				0.359
- G1	1 (2%)	0 (0%)	1 (2%)	
- G2	33 (58%)	6 (75%)	27 (55%)	
- G3	8 (14%)	2 (25%)	6 (12%)	
- Absent	12 (21%)	0 (0%)	12 (25%)	
- n/a	3 (5%)	0 (0%)	3 (6%)	

* explant features of patients transplanted for HCV-related HCC were available in 57/65 patients with HCC before LT. ** explant features of patients with HCC recurrence were available in 8/10 patients with HCC recurrence after LT. Among these patients, 2 had recurrent HCC before DAA treatment. *** explant features of patients with HCC before LT and no HCC recurrence were available in 49/55 patients.

## Data Availability

All data supporting the findings in the study are present within the article and are available upon reasonable request.

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
