# Peer review of "Long-Term Impact of Direct-Acting Antivirals on Liver Fibrosis and Survival in HCV-Infected Liver Transplant Recipients"

_viruses, 2023, doi:10.3390/v15081702_

Round 1

Reviewer 1 Report

The interesting article deals with the regression of advanced liver fibrosis and the associated reduction in the risk of hepatocellular carcinoma. It is highly significant and engaging, providing valuable insights from both a scientific and clinical perspective.

It has long been known that achieving SVR improves transplant patients' prognosis and actively replicates hepatitis C virus.

Current antiviral therapies use pan-genotypic drugs, but more relevant to this work is the effect of SVR on fibrosis regression, regardless of historical treatment regimens in this case.

I have a few questions/observations about the work to include in the article.

1.   Patients were observed until Jan 2019. Why don’t the antiviral regimens in Table 1 include the pan-genotypic regimens available in the EU from 2016/2017, which would better illustrate the currently recommended post-transplant antiviral therapies? Were the therapies included the only ones available, as written on page 2, or was a particular group of patients selected for the work for some reason?

2.   Did the same person always perform the elastography? Certified and with extensive experience in conducting such a survey? 

Was the elastography test performed only once in each patient? The mean follow-up after DAA treatment was 51 months. Assessment of liver tissue stiffness measurement was performed on average!!! At 33 months after EoT, there was no single point for all treated patients? Is it possible to draw clear conclusions about fibrotic changes from such data? The article's title refers to the relationship between DAA treatment and fibrosis degree, so can the actual changes in fibrosis degree be illustrated in detail, especially for F3-F4 patients (Figure 1 details the distribution of fibrosis during the observation period)? Can these data be related to the period when the elastography study was performed? Based on the data presented, is it possible to answer the question of how many patients with advanced fibrosis have fibrosis regression, and does this reduce the risk of HCC?

3.   In Table 1, 45 out of 56 patients with F4 were classified as CH=P A and B. What about the rest?

4. page 7

- Among the factors associated with reduced patient survival, the natural progression of liver failure in patients who have not responded to antiviral treatment is worth noting. None of the patients (as shown in Table 1) had a baseline platelet count < 50000/mm3, an effect secondary to disease progression.

- which means a higher creatinine level if the baseline data show a maximum creatinine concentration in the indicated range of 1.12 mg/dl, corresponding to normal or very good renal function. The issue of severe kidney damage at baseline is also in the discussion????

- how was liver decompensation defined? How many patients had ascites? Encephalopathy? Bleeding from oesophageal varices?

- In Figure 2, it is worth looking at the time measure (months) in more detail.

5. p. 8. In Table 2, the authors refer to albumin concentrations. These values are not given in the baseline characterisation.

6. Was the recurrence of HCC related to patients with cirrhosis/advanced fibrosis? In patients with a history of HCC and low grade <F2 fibrosis at the time of diagnosis?

7. Page 8, Table 3, what happened to the two patients with the 21 Milan criteria: out?

Reviewer 2 Report

The writing of the manuscript in general is not the most appropriate. The methodology has some limitations that could completely alter the conclusions of the study. On the other hand, the study is not very new in terms of the benefits of DAAs in the population studied. If the study has been carried out in patients who have received a liver transplant, more clinical data typical of this type of patient is needed, such as prescribed immunosuppressants.

Line 120: “Patients with HCC before LT who received DAAs after LT were analyzed separately”. The meaning of this phrase is not well understood.

Line 72. “Baseline staging of liver disease was performed by liver biopsy or transient elastography (TE)”. What method to assess liver stiffness was used at follow-up? Why do the authors mention that they will also take biopsies and esophageal varices as criteria for cirrhosis? Later this does not appear in the rest of the manuscript and confuses the reader about the criteria chosen by the authors.

Line 75. It is necessary to review the cut-offs. How is a patient classified with an LS of 8.5 kPa?

Line 138. “From the analysis: 2 patients for treatment discontinuation and 5 patients for insufficient follow-up to determine SVR12”. The meaning of this phrase is not well understood. It is assumed that the authors are talking about patients who were not included in the study. Later, the authors mention that 143 patients were included in the study, however, they again mention that there are 7 patients who were excluded. This wording confuses the reader.

According to Table 1, “LT indication”, there were 134 transplant patients. What happened to the other 2 lost patients?

Line 134, please provide the exact number of patients who have achieved SVR. 78% of patients suppose that 106.08 patients have achieved SVR.

Line 187, The improvement or worsening of liver stiffness may be due to the effect of follow-up time. Authors should also indicate the time between baseline and EoT. The authors only give the overall median and it would also be interesting to know the time elapsed in the “fibrosis improvement”, “unchaged liver fibrosis stage” and “fibrosis worsening” groups. In this sense, do patients who have worsened have more or less time elapsed between baseline and EoT than patients who have improved?

Line 347, “This is the only study evaluating the long-term impact of DAAs on liver fibrosis and survival in HCV-infected liver transplant recipients”. Please, I suggest authors review the scientific literature.

Reviewer 3 Report

This study aimed to evaluate the long-term impact of viral eradication on fibrosis progression and patient survival after LT in HCV-infected recipients. 

Several suggestions:

1.      Lines 69-70, [real-time polymerase chain reaction (RT-PCR)] is for the quantitation of DNA, please change to [real-time reverse transcriptase-polymerase chain reaction (real-time RT-PCR)].

2.      Lines 91-92, please check [National 21 and European guidelines 22.]?

3.      Line 105, [quantitative real-time PCR] should be [quantitative RT-PCR] or [real-time RT-PCR]. [quantitative] is the same meaning as [real-time].

4.      Lines 184-185, please add [LSM] after [liver stiffness measurements].

5.      Table 2, it is suggested to add [Hazard Ratio] for [HR].

6.      Line 257, please add references after [in LT recipients].

7.      Line 265, please add references after [in patients who received RBV].

8.      Lines 299-302, please check this sentence [The only study [26] which compared liver biopsies performed at baseline and after antiviral treatment within a long-term follow-up (3 years) included 12 patients with baseline liver cirrhosis and confirmed our result, demonstrating a significant improvement in hepatic fibrosis and inflammation.].

9.      Line 338, please add references after [in line with previous findings].
